# ExtremeWeather: A large-scale climate dataset for semi-supervised detection, localization, and understanding of extreme weather events

**Evan Racah**[1,2]**, Christopher Beckham**[1,3]**, Tegan Maharaj**[1,3]**,**
**Samira Ebrahimi Kahou**[4]**, Prabhat**[2]**, Christopher Pal**[1,3]
[1] MILA, Université de Montréal, `evan.racah@umontreal.ca`.
[2] Lawrence Berkeley National Lab, Berkeley, CA, `prabhat@lbl.gov`.
[3] École Polytechnique de Montréal, `firstname.lastname@polymtl.ca`.
[4] Microsoft Maluuba, `samira.ebrahimi@microsoft.com`.

## Abstract

Then detection and identification of extreme weather events in large-scale climate simulations is an important problem for risk management, informing governmental policy decisions and advancing our basic understanding of the climate system. Recent work has shown that fully supervised convolutional neural networks (CNNs) can yield acceptable accuracy for classifying well-known types of extreme weather events when large amounts of labeled data are available. However, many different types of spatially localized climate patterns are of interest including hurricanes, extra-tropical cyclones, weather fronts, and blocking events among others. Existing labeled data for these patterns can be incomplete in various ways, such as covering only certain years or geographic areas and having false negatives. This type of climate data therefore poses a number of interesting machine learning challenges. We present a multichannel spatiotemporal CNN architecture for semi-supervised bounding box prediction and exploratory data analysis. We demonstrate that our approach is able to leverage temporal information and unlabeled data to improve the localization of extreme weather events. Further, we explore the representations learned by our model in order to better understand this important data. We present a dataset, ExtremeWeather, to encourage machine learning research in this area and to help facilitate further work in understanding and mitigating the effects of climate change. The dataset is available at `extremeweatherdataset.github.io` and the code is available at `https://github.com/eracah/hur-detect`.

## 1  Introduction

Climate change is one of the most important challenges facing humanity in the 21st century, and climate simulations are one of the only viable mechanisms for understanding the future impact of various carbon emission scenarios and intervention strategies. Large climate simulations produce massive datasets: a simulation of 27 years from a 25 square km, 3 hour resolution model produces on the order of 10TB of multi-variate data. This scale of data makes post-processing and quantitative assessment challenging, and as a result, climate analysts and policy makers typically take global and annual averages of temperature or sea-level rise. While these coarse measurements are useful for public and media consumption, they ignore spatially (and temporally) resolved extreme weather events such as extra-tropical cyclones and tropical cyclones (hurricanes). Because the general public and policy makers are concerned about the *local* impacts of climate change, it is critical that we be able to examine how localized weather patterns (such as tropical cyclones), which can have dramatic impacts on populations and economies, will change in frequency and intensity under global warming.

Deep neural networks, especially deep convolutional neural networks, have enjoyed breakthrough success in recent recent years, achieving state-of-the-art results on many benchmark datasets (Krizhevsky et al., 2012; He et al., 2015; Szegedy et al., 2015) and also compelling results on many practical tasks such as disease diagnosis (Hosseini-Asl et al., 2016), facial recognition (Parkhi et al., 2015), autonomous driving (Chen et al., 2015), and many others. Furthermore, deep neural networks have also been very effective in the context of unsupervised and semi-supervised learning; some recent examples include variational autoencoders (Kingma & Welling, 2013), adversarial networks (Goodfellow et al., 2014; Makhzani et al., 2015; Salimans et al., 2016; Springenberg, 2015), ladder networks (Rasmus et al., 2015) and stacked what-where autoencoders (Zhao et al., 2015).

There is a recent trend towards video datasets aimed at better understanding spatiotemporal relations and multimodal inputs (Kay et al., 2017; Gu et al., 2017; Goyal et al., 2017). The task of finding extreme weather events in climate data is similar to the task of detecting objects and activities in video - a popular application for deep learning techniques. An important difference is that in the case of climate data, the 'video' has 16 or more 'channels' of information (such as water vapour, pressure and temperature), while conventional video only has 3 (RGB). In addition, climate simulations do not share the same statistics as natural images. As a result, unlike many popular techniques for video, we hypothesize that we cannot build off successes from the computer vision community such as using pretrained weights from CNNs (Simonyan & Zisserman, 2014; Krizhevsky et al., 2012) pretrained on ImageNet (Russakovsky et al., 2015).

Climate data thus poses a number of interesting machine learning problems: multi-class classification with unbalanced classes; partial annotation; anomaly detection; distributional shift and bias correction; spatial, temporal, and spatiotemporal relationships at widely varying scales; relationships between variables that are not fully understood; issues of data and computational efficiency; opportunities for semi-supervised and generative models; and more. Here, we address multi-class detection and localization of four extreme weather phenomena: tropical cyclones, extra-tropical cyclones, tropical depressions, and atmospheric rivers. We implement a 3D (height, width, time) convolutional encoder-decoder, with a novel single-pass bounding-box regression loss applied at the bottleneck. To our knowledge, this is the first use of a deep autoencoding architecture for bounding-box regression. This architectural choice allows us to do semi-supervised learning in a very natural way (simply training the autoencoder with reconstruction for unlabelled data), while providing relatively interpretable features at the bottleneck. This is appealing for use in the climate community, as current engineered heuristics do not perform as well as human experts for identifying extreme weather events.

Our main contributions are (1) a baseline bounding-box loss formulation; (2) our architecture, a first step away from engineered heuristics for extreme weather events, towards semi-supervised learned features; (3) the ExtremeWeather dataset, which we make available in three benchmarking splits: one small, for model exploration, one medium, and one comprising the full 27 years of climate simulation output.

## 2 Related work

### 2.1 Deep learning for climate and weather data

Climate scientists do use basic machine learning techniques, for example PCA analysis for dimensionality reduction (Monahan et al., 2009), and k-means analysis for clusterings Steinhaeuser et al. (2011). However, the climate science community primarily relies on expert engineered systems and ad-hoc rules for characterizing climate and weather patterns. Of particular relevance is the TECA (Toolkit for Extreme Climate Analysis) Prabhat et al. (2012, 2015), an application of large scale pattern detection on climate data using heuristic methods. A more detailed explanation of how TECA works is described in section 3. Using the output of TECA analysis (centers of storms and bounding boxes around these centers) as ground truth, (Liu et al., 2016) demonstrated for the first time that convolutional architectures could be successfully applied to predict the class label for two extreme weather event types. Their work considered the binary classification task on centered, cropped patches from 2D (single-timestep) multi-channel images. Like (Liu et al., 2016) we use TECA's output (centers and bounding boxes) as ground truth, but we build on the work of Liu et al. (2016) by: 1) using uncropped images, 2) considering the temporal axis of the data 3) doing multi-class bounding box detection and 4) taking a semi-supervised approach with a hybrid predictive and reconstructive model.

Some recent work has applied deep learning methods to weather forecasting. Xingjian et al. (2015) have explored a convolutional LSTM architecture (described in 2.2 for predicting future precipitation on a local scale (i.e. the size of a city) using radar echo data. In contrast, we focus on extreme event detection on planetary-scale data. Our aim is to capture patterns which are very local in time (e.g. a hurricane may be present in half a dozen sequential frames), compared to the scale of our underlying climate data, consisting of global simulations over many years. As such, 3D CNNs seemed to make more sense for our detection application, compared to LSTMs whose strength is in capturing long-term dependencies.

## 2.2 Related methods and models

Following the dramatic success of CNNs in static 2D images, a wide variety of CNN architectures have been explored for video, ex. (Karpathy et al., 2014; Yao et al., 2015; Tran et al., 2014). The details of how CNNs are extended to capture the temporal dimension are important. Karpathy et al. (2014) explore different strategies for fusing information from 2D CNN subcomponents; in contrast, Yao et al. (2015) create 3D volumes of statistics from low level image features.

Convolutional networks have also been combined with RNNs (recurrent neural networks) for modeling video and other sequence data, and we briefly review some relevant video models here. The most common and straightforward approach to modeling sequential images is to feed single-frame representations from a CNN at each timestep to an RNN. This approach has been examined for a number of different types of video (Donahue et al., 2015; Ebrahimi Kahou et al., 2015), while (Srivastava et al., 2015) have explored an LSTM architecture for the unsupervised learning of video representations using a pretrained CNN representation as input. These architectures separate learning of spatial and temporal features, something which is not desirable for climate patterns. Another popular model, also used on 1D data, is a convolutional RNN, wherein the hidden-to-hidden transition layer is 1D convolutional (i.e. the state is convolved over time). (Ballas et al., 2016) combine these ideas, applying a convolutional RNN to frames processed by a (2D) CNN.

The 3D CNNs we use here are based on 3-dimensional convolutional filters, taking the height, width, and time axes into account for each feature map, as opposed to aggregated 2D CNNs. This approach was studied in detail in (Tran et al., 2014). 3D convolutional neural networks have been used for various tasks ranging from human activity recognition (Ji et al., 2013), to large-scale YouTube video classification (Karpathy et al., 2014), and video description (Yao et al., 2015). Hosseini-Asl et al. (2016) use a 3D convolutional autoencoder for diagnosing Alzheimer's disease through MRI - in their case, the 3 dimensions are height, width, and depth. (Whitney et al., 2016) use 3D (height, width, depth) filters to predict consecutive frames of a video game for continuation learning. Recent work has also examined ways to use CNNs to generate animated textures and sounds (Xie et al., 2016). This work is similar to our approach in using 3D convolutional encoder, but where their approach is stochastic and used for generation, ours is deterministic, used for multi-class detection and localization, and also comprises a 3D convolutional decoder for unsupervised learning.

Stepping back, our approach is related conceptually to (Misra et al., 2015), who use semi-supervised learning for bounding-box detection, but their approach uses iterative heuristics with a support vector machine (SVM) classifer, an approach which would not allow learning of spatiotemporal features. Our setup is also similar to recent work from (Zhang et al., 2016) (and others) in using a hybrid prediction and autoencoder loss. This strategy has not, to our knowledge, been applied either to multidimensional data or bounding-box prediction, as we do here. Our bounding-box prediction loss is inspired by (Redmon et al., 2015), an approach extended in (Ren et al., 2015), as well as the single shot multiBox detector formulation used in (Liu et al., 2015) and the seminal bounding-box work in OverFeat (Sermanet et al., 2013). Details of this loss are described in Section 4.

## 3 The ExtremeWeather dataset

### 3.1 The Data

The climate science community uses three flavors of global datasets: observational products (satellite, gridded weather station); reanalysis products (obtained by assimilating disparate observational products into a climate model) and simulation products. In this study, we analyze output from the third category because we are interested in climate change projection studies. We would like to better understand how Earth's climate will change by the year 2100; and it is only possible to conduct

such an analysis on simulation output. Although this dataset contains the past, the performance of deep learning methods on this dataset can still inform the effectiveness of these approaches on future simulations. We consider the CAM5 (Community Atmospheric Model v5) simulation, which is a standardized three-dimensional, physical model of the atmosphere used by the climate community to simulate the global climate (Conley et al., 2012). When it is configured at 25-km spatial resolution (Wehner et al., 2015), each snapshot of the global atmospheric state in the CAM5 model output is a 768x1152 image, having 16 'channels', each corresponding to a different simulated variable (like surface temperature, surface pressure, precipitation, zonal wind, meridional wind, humidity, cloud fraction, water vapor, etc.). The global climate is simulated at a temporal resolution of 3 hours, giving 8 snapshots (images) per day. The data we provide is from a simulation of 27 years from 1979 to 2005. In total, this gives 78,840 16-channel 768x1152 images.

## 3.2 The Labels

Ground-truth labels are created for four extreme weather events: Tropical Depressions (TD) Tropical Cyclones (TC), Extra-Tropical Cyclones (ETC) and Atmospheric Rivers (AR) using TECA (Prabhat et al., 2012). TECA generally works by suggesting candidate coordinates for storm centers by only selecting points that follow a certain combination of criteria, which usually involves requiring various variables' (such as pressure, temperature and wind speed) values are between between certain thresholds. These candidates are then refined by breaking ties and matching the "same" storms across time (Prabhat et al., 2012). These storm centers are then used as the center coordinates for bounding boxes. The size of the boxes is determined using prior domain knowledge as to how big these storms usually are, as described in (Liu et al., 2016). Every other image (i.e. 4 per day) is labeled due to certain design decisions made during the production run of the TECA code. This gives us 39,420 labeled images.

### 3.2.1 Issues with the Labels

TECA, the ground truth labeling framework, implements heuristics to assign 'ground truth' labels for the four types of extreme weather events. However, it is entirely possible there are errors in the labeling: for instance, there is little agreement in the climate community on a standard heuristic for capturing Extra-Tropical Cyclones (Neu et al., 2013); Atmospheric Rivers have been extensively studied in the northern hemisphere (Lavers et al., 2012; Dettinger et al., 2011), but not in the southern hemisphere; and spatial extents of such events not universally agreed upon. In addition, this labeling only includes AR's in the US and not in Europe. As such, there is potential for many false negatives, resulting in partially annotated images. Lastly, it is worth mentioning that because the ground truth generation is a simple automated method, a deep, supervised method can only do as well as emulating this class of simple functions. This, in addition to lower representation for some classes (AR and TD), is part of our motivation in exploring semi-supervised methods to better understand the features underlying extreme weather events rather than trying to "beat" existing techniques.

## 3.3 Suggested Train/Test Splits

We provide suggested train/test splits for the varying sizes of datasets on which we run experiments. Table 1 shows the years used for train and test for each dataset size. We show "small" (2 years train, 1 year of test), "medium" (8 years train, 2 years test) and "large" (22 years train, 5 years test) datasets. For reference, table 2 shows the breakdown of the dataset splits for each class for "small" in order to illustrate the class-imbalance present in the dataset. Our model was trained on "small", where we split the train set 50:50 for train and validation. Links for downloading train and test data, as well as further information the different dataset sizes and splits can be found here: `extremeweatherdataset.github.io`.

Table 1: Three benchmarking levels for the ExtremeWeather dataset

| Level | Train | Test |
|-------|-------|------|
| Small | 1979, 1981 | 1984 |
| Medium | 1979-1983,1989-1991 | 1984-1985 |
| Large | 1979-1983, 1994-2005, 1989-1993 | 1984-1988 |

Table 2: Number of examples in ExtremeWeather benchmark splits, with class breakdown statistics for Tropical Cyclones (**TC**), Extra-Tropical Cyclones (**ETC**), Tropical Depressions (**TD**), and United States Atmospheric Rivers (**US-AR**)

| Benchmark | Split | TC (%) | ETC (%) | TD (%) | US-AR (%) | Total |
|-----------|-------|--------|---------|--------|-----------|-------|
| Small | Train | 3190 (42.32) | 3510 (46.57) | 433 (5.74) | 404 (5.36) | 7537 |
|  | Test | 2882 (39.04) | 3430 (46.47) | 697 (9.44) | 372 (5.04) | 7381 |

## 4  The model

We use a 3D convolutional encoder-decoder architecture, meaning that the filters of the convolutional encoder and decoder are 3 dimensional (height, width, time). The architecture is shown in Figure 1; the encoder uses convolution at each layer while the decoder is the equivalent structure in reverse, using tied weights and deconvolutional layers, with leaky ReLUs (Andrew L. Maas & Ng., 2013) (0.1) after each layer. As we take a semi-supervised approach, the code (bottleneck) layer of the autoencoder is used as the input to the loss layers, which make predictions for (1) bounding box location and size, (2) class associated with the bounding box, and (3) the confidence (sometimes called 'objectness') of the bounding box. Further details (filter size, stride length, padding, output sizes, etc.) can be found in the supplementary materials.

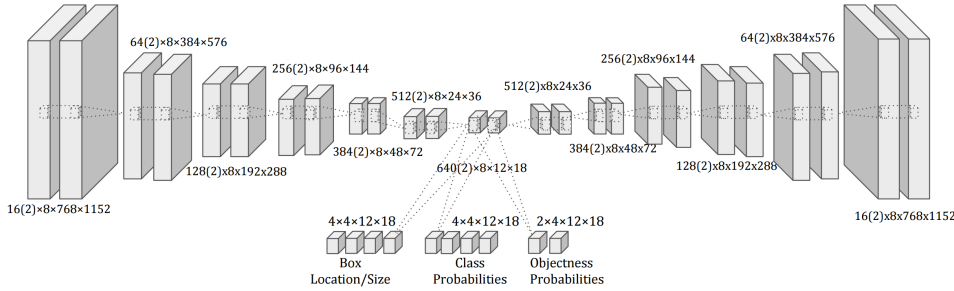

Figure 1: Diagram of the 3D semi-supervised architecture. Parentheses denote subset of total dimension shown (for ease of visualization, only two feature maps per layer are shown for the encoder-decoder. All feature maps are shown for bounding-box regression layers).

The total loss for the network, $L$, is a weighted combination of supervised bounding-box regression loss, $L_{sup}$, and unsupervised reconstruction error, $L_{rec}$:

$$L = L_{sup} + \lambda L_{rec}, \tag{1}$$

where $L_{rec}$ is the mean squared squared difference between input $X$ and reconstruction $X^*$:

$$L_{rec} = \frac{1}{M}||X - X^*||_2^2, \tag{2}$$

where M is the total number of pixels in an image.

In order to regress bounding boxes, we split the original 768x1152 image into a 12x18 grid of 64x64 anchor boxes. We then predict a box at each grid point by transforming the representation to 12x18=216 scores (one per anchor box). Each score encodes three pieces of information: (1) how much the predicted box differs in size and location from the anchor box, (2) the confidence that an object of interest is in the predicted box ("objectness"), and (3) the class probability distribution for that object. Each component of the score is computed by several 3x3 convolutions applied to the 640 12x18 feature maps of the last encoder layer. Because each set of pixels in each feature map at a given $x, y$ coordinate can be thought of as a learned representation of the climate data in a 64x64 patch of the input image, we can think of the 3x3 convolutions as having a local receptive field size of 192x192, so they use a representation of a 192x192 neighborhood from the input image as context to determine the box and object centered in the given 64x64 patch. Our approach is similar to (Liu et al., 2015) and (Sermanet et al., 2013), which use convolutions from small local receptive field filters to

regress boxes. This choice is motivated by the fact that extreme weather events occur in relatively small spatiotemporal volumes, with the 'background' context being highly consistent across event types and between events and non-events. This is in contrast to Redmon et al. (2015), which uses a fully connected layer to consider the whole image as context, appropriate for the task of object identification in natural images, where there is often a strong relationship between background and object.

The bounding box regression loss, $L_{sup}$, is determined as follows:

$$L_{sup} = \frac{1}{N}(L_{box} + L_{conf} + L_{cls}), \tag{3}$$

where N is the number of time steps in the minibatch, and $L_{box}$ is defined as:

$$L_{box} = \alpha \sum_i \mathbb{1}_i^{obj} R(u_i - u_i^*) + \beta \sum_i \mathbb{1}_i^{obj} R(v_i - v_i^*), \tag{4}$$

where $i \in [0, 216)$ is the index of the anchor box for the ith grid point, and where $\mathbb{1}_i^{obj} = 1$ if an object is present at the ith grid point, 0 if not; $R(z)$ is the smooth L1 loss as used in (Ren et al., 2015), $u_i = (t_x, t_y)_i$ and $u_i^* = (t_x^*, t_y^*)_i$, $v_i = (t_w, t_h)_i$ and $v_i^* = (t_w^*, t_h^*)_i$ and $t$ is the parametrization defined in (Ren et al., 2015) such that:

$$t_x = (x - x_a)/w_a, t_y = (y - y_a)/h_a, t_w = \log(w/w_a), t_h = \log(h/h_a)$$

$$t_x^* = (x^* - x_a)/w_a, t_y^* = (y^* - y_a)/h_a, t_w^* = \log(w^*/w_a), t_h^* = \log(h^*/h_a),$$

where $(x_a, y_a, w_a, h_a)$ is the center coordinates and height and width of the closest anchor box, $(x, y, w, h)$ are the predicted coordinates and $(x^*, y^*, w^*, h^*)$ are the ground truth coordinates.

$L_{conf}$ is the weighted cross-entropy of the log-probability of an object being present in a grid cell:

$$L_{conf} = \sum_i \mathbb{1}_i^{obj}[-\log(p(obj)_i)] + \gamma * \sum_i \mathbb{1}_i^{noobj}[-\log(p(\overline{obj}_i))] \tag{5}$$

Finally $L_{cls}$ is the cross-entropy between the one-hot encoded class distribution and the softmax predicted class distribution, evaluated only for predicted boxes at the grid points containing a ground truth box:

$$L_{cls} = \sum_i \mathbb{1}_i^{obj} \sum_{c \in classes} -p^*(c) \log(p(c)) \tag{6}$$

The formulation of $L_{sup}$ is similar in spirit to YOLO (Redmon et al., 2015), with a few important differences. Firstly, the object confidence and class probability terms in YOLO are squared-differences between ground truth and prediction, while we use cross-entropy, as used in the region proposal network from Faster R-CNN (Ren et al., 2015) and the network from (Liu et al., 2015), for the object probability term and the class probability term respectively. Secondly, we use a different parametrization for the coordinates and the size of the bounding box. In YOLO, the parametrizations for $x$ and $y$ are equivalent to Faster R-CNN's $t_x$ and $t_y$, for an anchor box the same size as the patch it represents (64x64). However $w$ and $h$ in YOLO are equivalent to Faster-RCNN's $t_h$ and $t_w$ for a 64x64 anchor box only if (a) the anchor box had a height and width equal to the size of the whole image and (b) there were no log transform in the faster-RCNN's parametrization. We find both these differences to be important in practice. Without the log term, and using ReLU nonlinearities initialized (as is standard) centered around 0, most outputs (more than half) will give initial boxes that are in 0 height and width. This makes learning very slow, as the network must learn to resize essentially empty boxes. Adding the log term alone in effect makes the "default" box (an output of 0) equal to the height and width of the entire image - this equally slows down learning, because the network must now learn to drastically shrink boxes. Making $h_a$ and $w_a$ equal to 64x64 is a pragmatic 'Goldilocks' value. This makes training much more efficient, as optimization can focus more on picking *which* box contains an object and not as much on what size the box should be. Finally, where YOLO uses squared difference between predicted and ground truth for the coordinate parametrizations, we use smooth L1, due its lower sensitivity to outlier predictions (Ren et al., 2015).

# 5 Experiments and Discussion

## 5.1 Framewise Reconstruction

As a simple experiment, we first train a 2D convolutional autoencoder on the data, treating each timestep as an individual training example (everything else about the model is as described in Section 4), in order to visually assess reconstructions and ensure reasonable accuracy of detection. Figure 2 shows the original and reconstructed feature maps for the 16 climate variables of one image in the training set. Reconstruction loss on the validation set was similar to the training set. As the reconstruction visualizations suggest, the convolutional autoencoder architecture does a good job of encoding spatial information from climate images.

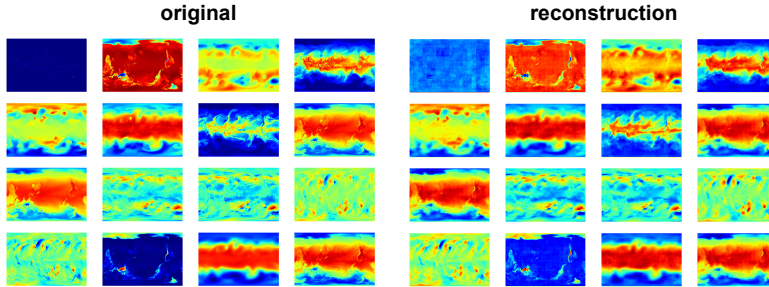

Figure 2: Feature maps for the 16 channels in an 'image' from the training set (left) and their reconstructions from the 2D convolutional autoencoder (right).

## 5.2 Detection and localization

All experiments are on ExtremeWeather-small, as described in Section 3, where 1979 is train and 1981 is validation. The model is trained with Adam (Kingma & Ba, 2014), with a learning rate of 0.0001 and weight decay coefficient of 0.0005. For comparison, and to evaluate how useful the time axis is to recognizing extreme weather events, we run experiments with both **2D** (width, height) and **3D** (width, height, time) versions of the architecture described in Section 4. Values for $\alpha, \beta, \gamma$ (hyperparameters described in loss Equations 4 and 5) were selected with experimentation and some inspiration from (Redmon et al., 2015) to be 5, 7 and 0.5 respectively. A lower value for $\gamma$ pushes up the confidence of true positive examples, allowing the model more examples to learn from, is thus a way to deal with ground-truth false negatives. Although some of the selection of these parameters is a bit ad-hoc, we assert that our results still provide a good first-pass baseline approach for this dataset. The code is available at `https://github.com/eracah/hur-detect`

During training, we input one day's simulation at a time (8 time steps; 16 variables). The **semi-supervised** experiments reconstruct all 8 time steps, predicting bounding boxes for the 4 labelled timesteps, while the **supervised** experiments reconstruct and predict bounding boxes only for the 4 labelled timesteps. Table 3 shows Mean Average Precision (mAP) for each experiment. Average Precision (AP) is calculated for each class in the manner of ImageNet (Russakovsky et al., 2015), integrating the precision-recall curve, and mAP is averaged over classes. Results are shown for various settings of $\lambda$ (see Equation 1) and for two modes of evaluation; at IOU (intersection over union of the bounding-box and ground-truth box) thresholds of 0.1 and 0.5. Because the 3D model has inherently higher capacity (in terms of number of parameters) than the 2D model, we also experiment with higher capacity 2D models by doubling the number of filters in each layer. Figure 3 shows bounding box predictions for 2 consecutive (6 hours in between) simulation frames, comparing the 3D supervised vs 3D semi-supervised model predictions.

It is interesting to note that 3D models perform significantly better than their 2D counterparts for ETC and TC (hurricane) classes. This implies that the time evolution of these weather events is an important criteria for discriminating them. In addition, the semi-supervised model significantly improves the ETC and TC performance, which suggests unsupervised shaping of the spatio-temporal representation is important for these events. Similarly, semi-supervised data improves performance of the 3D model (for IOU=0.1), while this effect is not observed for 2D models, suggesting that 3D representations benefit more from unsupervised data. Note that hyperparameters were tuned in the supervised setting, and a more thorough hyperparameter search for $\lambda$ and other parameters may yield better semi-supervised results.

Figure 3 shows qualitatively what the quantitative results in Table 3 confirm - semi-supervised approaches help with rough localization of weather events, but the model struggles to achieve accurate boxes. As mentioned in Section 4, the network has a hard time adjusting the size of the boxes. As such, in this figure we see mostly boxes of size 64x64. For example, for TDs (usually much smaller than 64x64) and for ARs, (always much bigger than 64x64), a 64x64 box roughly centered on the event is sufficient to count as a true positive at IOU=0.1, but not at the more stringent IOU=0.5. This lead to a large dropoff in performance for ARs and TDs, and a sizable dropoff in the (variably-sized) TCs. Longer training time could potentially help address these issues.

Table 3: 2D and 3D supervised and semi-supervised results, showing Mean Average Precision (mAP) and Average Precision (AP) for each class, at IOU=0.1; IOU=0.5. **M** is model; **P** is millions of parameters; and $\lambda$ weights the amount that reconstruction contributes to the overall loss.

| M | Mode | P | $\lambda$ | ETC (46.47%) AP (%) | TC (39.04%) AP (%) | TD (9.44%) AP (%) | AR (5.04%) AP (%) | mAP |
|---|------|---|---|---------------------|--------------------|-------------------|-------------------|-----|
| 2D | Sup | 66.53 | 0 | 21.92; 14.42 | 52.26; 9.23 | 95.91; 10.76 | 35.61; 33.51 | 51.42; **16.98** |
| 2D | Semi | 66.53 | 1 | 18.05; 5.00 | 52.37; 5.26 | 97.69; 14.60 | 36.33; 0.00 | 51.11; 6.21 |
| 2D | Semi | 66.53 | 10 | 15.57; 5.87 | 44.22; 2.53 | 98.99; **28.56** | **36.61**; 0.00 | 48.85; 9.24 |
| 2D | Sup | 16.68 | 0 | 13.90; 5.25 | 49.74; **15.33** | 97.58; 7.56 | 35.63; **33.84** | 49.21; 15.49 |
| 2D | Semi | 16.68 | 1 | 15.80; 9.62 | 39.49; 4.84 | **99.50**; 3.26 | 21.26; 13.12 | 44.01; 7.71 |
| 3D | Sup | 50.02 | 0 | 22.65; **15.53** | 50.01; 9.12 | 97.31; 3.81 | 34.05; 17.94 | 51.00; 11.60 |
| 3D | Semi | 50.02 | 1 | **24.74**; 14.46 | **56.40**; 9.00 | 96.57; 5.80 | 33.95; 0.00 | **52.92**; 7.31 |

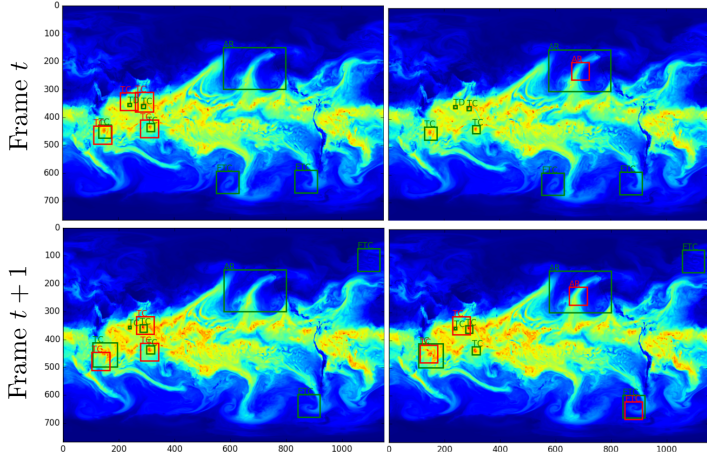

Figure 3: Bounding box predictions shown on 2 consecutive (6 hours in between) simulation frames, for the integrated water vapor column channel. Green = ground truth, Red = high confidence predictions (confidence above 0.8). 3D supervised model (Left), and semi-supervised (Right).

### 5.3 Feature exploration

In order to explore learned representations, we use t-SNE (van der Maaten & Hinton, Nov 2008) to visualize the autoencoder bottleneck (last encoder layer). Figure 4 shows the projected feature maps for the first 7 days in the training set for both 3D supervised (top) and semi-supervised (bottom) experiments. Comparing the two, it appears that more TCs (hurricanes) are clustered by the semi-supervised model, which would fit with the result that semi-supervised information is particularly valuable for this class. Viewing the feature maps, we can see that both models have learned spiral patterns for TCs and ETCs.

## 6 Conclusions and Future Work

We introduce to the community the ExtremeWeather dataset in hopes of encouraging new research into unique, difficult, and socially and scientifically important datasets. We also present a baseline method for comparison on this new dataset. The baseline explores semi-supervised methods for

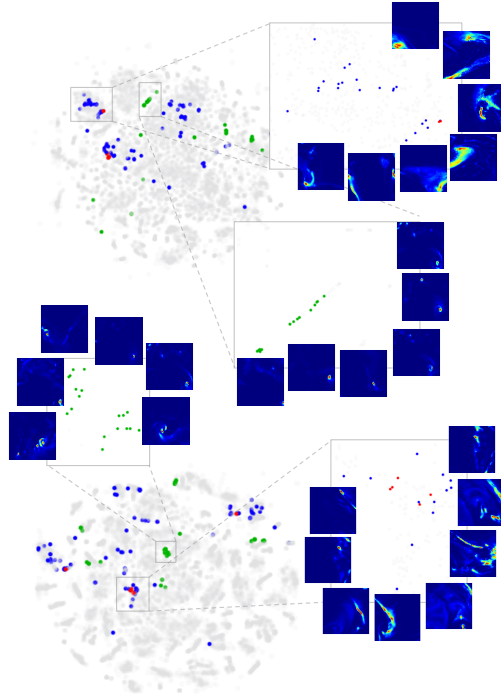

Figure 4: t-SNE visualisation of the first 7 days in the training set for 3D supervised (top) and semi-supervised (bottom) experiments. Each frame (time step) in the 7 days has 12x18 = 216 vectors of length 640 (number of feature maps in the code layer), where each pixel in the 12x18 patch corresponds to a 64x64 patch in the original frame. These vectors are projected by t-SNE to two dimensions. For both supervised and semi-supervised, we have zoomed into two dense clusters and sampled 64x64 patches to show what that feature map has learned. Grey = unlabelled, Yellow = tropical depression (not shown), Green = TC (hurricane), Blue = ETC, Red = AR.

object detection and bounding box prediction using 3D autoencoding CNNs. These architectures and approaches are motivated by finding extreme weather patterns; a meaningful and important problem for society. Thus far, the climate science community has used hand-engineered criteria to characterize patterns. Our results indicate that there is much promise in considering deep learning based approaches. Future work will investigate ways to improve bounding-box accuracy, although even rough localizations can be very useful as a data exploration tool, or initial step in a larger decision-making system. Further interpretation and visualization of learned features could lead to better heuristics, and understanding of the way different variables contribute to extreme weather events. Insights in this paper come from only a fraction of the available data, and we have not explored such challenging topics as anomaly detection, partial annotation detection and transfer learning (e.g. to satellite imagery). Moreover, learning to generate future frames using GAN's (Goodfellow et al., 2014) or other deep generative models, while using performance on a detection model to measure the quality of the generated frames could be another very interesting future direction. We make the ExtremeWeather dataset available in hopes of enabling and encouraging the machine learning community to pursue these directions. The retirement of Imagenet this year (Russakovsky et al., 2017) marks the end of an era in deep learning and computer vision. We believe the era to come should be defined by data of social importance, pushing the boundaries of what we know how to model.

## Acknowledgments

This research used resources of the National Energy Research Scientific Computing Center (NERSC), a DOE Office of Science User Facility supported by the Office of Science of the U.S. Department of Energy under Contract No. DE-AC02-05CH11231. Code relies on open-source deep learning frameworks Theano (Bergstra et al.; Team et al., 2016) and Lasagne (Team, 2016), whose developers we gratefully acknowledge. We thank Samsung and Google for support that helped make this research

possible. We would also like to thank Yunjie Liu and Michael Wehner for providing access to the climate datasets; Alex Lamb and Thorsten Kurth for helpful discussions.

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
