[Reviews · NeurIPS 2017]

Reviewer 1



Climate change is a critically important area of research and modeling of spatially localized climate patterns can be imperative. In this work, the authors describe the creation of the ExtremeWeather dataset generated from a 27 year run of the CAM5 model. As configured, the output consists of a 768x1152 image with 16 channels. 4 extreme weather events have ground truth labeling generated using the TECA framework. ~79K images, about half of which are labeled are generated. A 3D convolutional encoder decoder architecture was employed. Only the surface quantities were modeled (3D, with 3rd dimension being time). Experiments were run in 2D (without the temporal aspecct) and 3D The connections between climate change and local weather patterns could be made more clear to audiences less familiar with the topic. The reason for using simulated data versus observations could also be articulated better. Not sure why only the surface level data was used in generating the dataset -one would think that the atmospheric data (30 levels) would have implications for the 4 extreme weather events that were modeled. In many cases, the choice of architecture/optimal parameters seems to be a little ad-hoc.

Reviewer 2



This paper presents a new dataset, a model and experimental results on this dataset to address the task of extreme weather events detection and localization. The dataset is 27 year weather simulation sampled 8 times per day for 16 channels only the surface atmospheric level. The proposed model is based on 3D convolutional layers with an autoencoder architecture. The technique is semi-supervised, thus training with a loss that combines reconstruction error of the autoencoder and detection and localization from the middle code layer. In general the paper is very well written and quite clear on most details. The experimental results only use a small part of the data and are a bit preliminary, but still they do show the potential that this data has for future research. Some comments/concerns/suggestions are the following. - Since the paper presents a new dataset it is important to include a link to where the data will be available. - In line 141 it says that only surface quantities are considered without any explanation why. Is it because the simulation would have been significantly more expensive so only surface computed? Is it a decision of the authors to make the dataset size more manageable? Are there any future plans to make available the full 30 atmospheric levels? - The caption of table 1 says "Note test statistics are omitted to preserve anonymity of the test data." I assume this means that the ground truth labels are not being made available with the rest of the data. If this is the case, in the paper it should be explained what is the plan for other researchers to evaluate the techniques they develop. Will there be some web server where results are submitted and the results are returned? - From what I understand the ground truth labels for the four weather events considered were generated fully automatically. If so, why was only half of the data labeled? I agree that exploring semi-supervised methods is important, but this can be studied by ignoring the labels of part of the data and analyze the behavior of the techniques when they have available more or less labeled samples. - Is there any plan to extend the data with weather events that cannot be labeled by TECA? - VGG is mentioned in line 51 without explaining what it is. Maybe add a reference for readers that are not familiar with it. - In figure 3 the the graphs have very tiny numbers. These seem useless so maybe remove them saving a bit of space to make the images slightly larger. - Line 172 final full stop missing. - Be consistent on the use of t-SNE, some places you have it with a dash and other places without a dash.

Reviewer 3



This paper introduces a climate dataset. For baseline performance, the authors present a multichannel spatiotemporal encoder-decoder CNN architecture for semi-supervised bounding box prediction and exploratory data analysis. The paper presents a 27-year run, from 1979 - 2005, of the CAM5 (Community Atmospheric Model v5), configured at 25-km spatial resolution, with ground- truth labelling for four extreme weather events: Tropical Depressions (TD) Tropical Cyclones (TC), Extra-Tropical Cyclones (ETC) and Atmospheric Rivers (AR). At this resolution, each snapshot of the global atmospheric state in the CAM5 model output corresponds to a 768x1152 image, having 16 ’channels’ (including surface temperature, surface pressure, precipitation, zonal wind, meridional, wind, humidity, cloud fraction, water vapor, etc.). The goal is a 4 class classification (ETC, TD, TC, AR). The paper alludes to the climate dataset being similar to a multi-channel video dataset. A greater need for video datasets that can better understand multimodal inputs, understand video relationships and also push the boundaries of deep learning research. Towards this goal, many new datasets such as Kinetics, AVA, "something something" have been recently proposed. I feel the related work can be expanded significantly. How did you obtain this dataset? Is this a public dataset from CAM5 community? You create a much smaller dataset. What prevents you from using a larger dataset? More details about the dataset collection process are necessary. Furthermore, if you are just focussing on the weather/climate community, discussing previous work to handle weather data will be useful (there is some hint in the conclusions). More importantly, additional simpler baselines such as using CNN, and other feature information such flow that is currently used for object and activity recognition in videos will be helpful. Overall, I am not convinced from the paper how this dataset pushes for a new class of video dataset. While I agree that climate research is important, it is unclear if this is a niche area/dataset and how improving the results of this dataset will broadly applicable to improving the state of . Finally, additional baselines will also benefit the paper.